# 3D Ultrasensitive Polymers-Plasmonic Hybrid Flexible Platform for In-Situ Detection

**DOI:** 10.3390/polym12020392

**Published:** 2020-02-09

**Authors:** Meimei Wu, Chao Zhang, Yihan Ji, Yuan Tian, Haonan Wei, Chonghui Li, Zhen Li, Tiying Zhu, Qianqian Sun, Baoyuan Man, Mei Liu

**Affiliations:** 1School of Physics and Electronics, Shandong Normal University, Jinan 250014, China; sdnumm@163.com (M.W.); czsdnu@126.com (C.Z.); Ji_Yi_Han@163.com (Y.J.); 2018020524@stu.sdnu.edu.cn (Y.T.); 2018020530@stu.sdnu.edu.cn (H.W.); chonghuili163@163.com (C.L.); lizhen19910528@163.com (Z.L.); 2018020534@stu.sdnu.edu.cn (T.Z.); qianqiansun@sdnu.edu.cn (Q.S.); byman@sdnu.edu.cn (B.M.); 2Institute of Materials and Clean Energy, Shandong Normal University, Jinan 250014, China

**Keywords:** SERS, PMMA, AgNPs, in-situ, adenosine, methylene-blue

## Abstract

This paper introduces a three-dimensional (3D) pyramid to the polymers-plasmonic hybrid structure of polymethyl methacrylate (PMMA) composite silver nanoparticle (AgNPs) as a higher quality flexible surface-enhanced Raman scattering (SERS) substrate. Benefiting from the effective oscillation of light inside the pyramid valley could provide wide distributions of 3D “hot spots” in a large space. The inclined surface design of the pyramid structure could facilitate the aggregation of probe molecules, which achieves highly sensitive detection of rhodamine 6G (R6G) and crystal violet (CV). In addition, the AgNPs and PMMA composite structures provide uniform space distribution for analyte detection in a designated hot spot zone. The incident light can penetrate the external PMMA film to trigger the localized plasmon resonance of the encapsulated AgNPs, achieving enormous enhancement factor (~$6.24\times 10^{8}$). After undergoes mechanical deformation, the flexible SERS substrate still maintains high mechanical stability, which was proved by experiment and theory. For practical applications, the prepared flexible SERS substrate is adapted to the in-situ Raman detection of adenosine aqueous solution and the methylene-blue (MB) molecule detection of the skin of a fish, providing a direct and nondestructive active-platform for the detecting on the surfaces with any arbitrary morphology and aqueous solution.

## 1. Introduction

With ultra-sensitive, rapid, and non-destructive properties, surface-enhanced Raman scattering (SERS) is generally considered a valuable analytical technique for achieving label-free detection of biochemical molecules [1,2,3]. Recent research concerning SERS active substrates focused on designing and fabricating metallic nanostructures with tunable plasmonic features on hard substrates [4,5,6,7]. Significant Raman enhancement tends to occur at nanogaps between neighboring metal nanostructures with high local-field intensities (so-called “hot spots”) [8,9,10]. However, in many cases, SERS substrates based on silicon wafers or glass slides are rigid, brittle, and hence unsuitable for directly analyzing target molecules in a solution or gas [11,12]. Compared to traditional rigid substrates, flexible substrates can wrap around complex surfaces and be tailored to the desired size and shape [13]. In addition to achieving a highly active SERS substrate, the effective and rapid collection of probe molecules to the substrate presents a significant challenge [12]. 

Generally, flexible substrates with simple and periodic two-dimensional (2D) structures are limited by their low density of hot spots distributed at the gaps between horizontal nanoparticals (NPs), which leads to weak Raman enhancement [14]. Thus, to further improve SERS sensitivity, the integration of flexible three-dimensional (3D) structures is imperative. Such 3D flexible SERS substrates can offer large specific surfaces and create high-density hot spots to capture more target molecules [15]. Recently, great efforts were dedicated to fabricating highly sensitive flexible substrates by combining noble plasmonic nanoparticles with polymers. Park et al. have demonstrated a transparent and flexible SERS substrate based on the polydimethylsiloxane (PDMS) film embedded with gold nanostar (GNS) [13]. Kumar et al. fabricated a 3D flexible SERS substrate by depositing AgNPs on structured polydimethylsiloxane using a Taro leaf as the template to detect malachite green [16]. Kumar et al. fabricated a buckled PDMS silver nanorods array as an active 3D SERS sensor for detecting bacteria [17]. Despite demonstrating enormous application prospects for real sample detection, these flexible substrates still involve unresolved issues regarding metal-molecule contact and metal particle oxidation, both of which can cause poor reproducibility and stability in SERS detection [17,18]. 

In this study, we propose a 3D pyramid-shaped AgNPs@PMMA flexible composite platform for SERS application. The inclined surface of the periodic pyramid-shaped structures could facilitate the aggregation of probe molecules and create high-density 3D “hot spots”. In addition, the AgNPs and PMMA composite structures provide uniform space distribution for analyte detection in designated hot spot zone, and the incident light can penetrate the external PMMA film to trigger the localized plasmon resonance of the encapsulated AgNPs, achieving enormous enhancement factor (~$6.24\times 10^{8}$). The fabricated 3D pyramid-shaped AgNPs@PMMA flexible substrate achieved highly sensitive detection of rhodamine 6G (R6G) and crystal violet (CV), and displayed both high stability and signal reproducibility. Following mechanical stimuli in our experiments, such as stretching and bending, the flexible substrate maintained the stable Raman signal, which corroborates our theoretical simulation. Finally, we successfully achieve the in-situ detection in adenosine aqueous solution and the methylene-blue (MB) molecule detection of the skin of a fish, which demonstrates its great potential for the detecting on the surfaces with any arbitrary morphology and aqueous solution.

## 2. Materials and Methods

### 2.1. Materials

Acetone (CH_3_COCH_3_, 99.5%), alcohol (C_2_H_6_O, 99.7%), ethylene glycol (C_2_H_6_O_2_, 99.0%), R6G, crystal violet (CV), MB, silver nitrate (AgNO_3_), and polymethyl methacrylate (PMMA) were all obtained from Sinopharm Chemical Reagent Co. Ltd. (Shanghai, China). Polyvinylpyrrolidone (PVP, *M*w = 55,000) was obtained from Sigma-Aldrich (St. Louis, MO, USA), and the adenosine sample was purchased from Sangon Biotech (Shanghai) Co. Ltd. (Shanghai, China). The adenosine was dissolved in ultrapure water to make a 0.01M stock solution and then diluted to the final concentration before use.

### 2.2. Fabrication of 3D Pyramid AgNPs @PMMA Flexible Substrate

Figure 1 schematically illustrates the fabrication procedure for the 3D-pyramid-shaped AgNPs @PMMA flexible (P–AgNPs@PMMA) substrate. Pyramid Si substrate (P–Si) was pretreated using the method of wet texturing boron-doped monocrystalline silicon in a NaOH solution [19]. AgNPs dispersion was synthesized by the oil bath synthesis, according to previous work [20]. First, 0.075 g of PMMA grains were put into 100 mL of acetone, then this solution was stirred at 80 °C for 1 h. High-density and uniform AgNPs dispersion was added into the PMMA solution, and subsequently, this mixed solution was stirred at room temperature and allowed to stand overnight. The mixed solution was then dip-coated on the P-Si substrate, after the mixed layer was dried, the sample was immersed in a NaOH (30%, 100 mL) solution for 1 h to remove the P–Si supporter. The flexible films were rinsed with running deionized water, inverted on a glass substrate, and dried at room temperature to be used as the SERS substrate in our experiments.

### 2.3. Characterization

Scanning electron microscopy (SEM) images of the P–AgNPs@PMMA substrate were characterized using a scanning electron microscope (SEM, Sigma 500, Carl Zeiss, Jena, Germany) with an energy dispersive spectrometer (EDS) at a voltage of 10 kV. The SERS spectra of the P–AgNPs@PMMA substrate were recorded by a Raman spectrometer (LabRAM HR Evolution, Horiba, Kyoto, Japan) with 532 nm laser excitation. The effective power of the laser source was kept at 0.048 mW, and a 50× objective (N.A. = 0.50) was used throughout the test. The diffraction grid was 600 gr/nm, and the integration time was 8 s. All the SERS spectra in the experiment are expressed on the basis of average spectra. The UV-visible absorption spectra of the synthesized silver nanoparticles was collected by a dual beam UV-visible spectrophotometer (TU-1900, Beijing General Analysis General Instrument Co., Ltd, Beijing, China).

## 3. Results and Discussion

To investigate the structural properties of the P–AgNPs@PMMA composite substrates, we performed SEM characterization (Figure 2). AgNPs were deposited on the surface of the P-Si substrate via dip-coating, as shown in Figure 2a. The inclined surface of the periodic pyramid-shaped structures could facilitate the aggregation of probe molecules and create high-density 3D “hot spots”, which is critical for the final enhancement activity. Figure 2b,c, respectively exhibit low- and high-magnification SEM images of the P–AgNPs@PMMA flexible substrate after removal of the P-Si substrate. As pictured in Appendix A (from the Appendix A), the average size of these AgNPs is ∼78nm, and the gaps among nanoparticles are very narrow, which supports huge electromagnetic enhancement for absorbed molecules. The composition of the P–AgNPs @ PMMA flexible substrate is also analyzed with high-resolution EDS mapping in Figure 2d,e, indicating that high-density AgNPs were successfully embedded within the PMMA film. Figure 2f shows a strong UV-vis absorption peak at ~476 nm of AgNPs, which is attributed to the strong coupling of the plasma between the AgNPs. Thus, a high signal-to-noise ratio SERS signal can be generated at 532 nm incident excitation light.

It has been reported that high-density “hot spots” are always generated at the slits or tips of metal nanostructures and the interspacing gaps between the adjacent metal nanostructures could markedly affect the ability of detection limit and the intensity of SERS signals [15]. To further optimize the Raman enhancement effect of the composite SERS substrate, we adjusted the distribution density of AgNPs in the PMMA solution to find the best PMMA/Ag volume ratio. Figure 3a shows the SERS spectra for R6G (10^−7^ M) absorbed on AgNPs@PMMA/P-Si substrates with different PMMA /Ag volume ratio. The characteristic SERS peaks of R6G were observed around 612, 773, 1308, 1358, 1505, and 1645 cm^−1^. The peak at 612 cm^−1^ is attributed to C–C–C bond stretching modes. The peak at 773 cm^−1^ is due to the R6G C−H out-of-plane bending mode. The peaks around 1308, 1358, 1505, and 1645 cm^−1^ originate from aromatic C–C stretching [21,22]. It is observed that the Raman signal of the R6G molecule shows an apparent trend from the rise to decline with the continuous addition of silver colloid in a volume of PMMA solution. From Figure 3b, we can see that the SERS performance of the substrate is optimal with the PMMA/Ag volume ratio of 2:3. As shown in Appendix A (from the Appendix A), the SEM images of different volume ratio AgNPs@PMMA/P-Si substrate further illustrate that the number of AgNPs is small and sparse in the initial stage of the reaction. The area where “hot spots” generated between the silver particles is small, and the Raman signal is weak. As silver particles become denser with the continuous addition of silver colloid, the particle spacing decreases, which causes the increases of the density of “hot spots”, and further the Raman signal of the molecule gradually increases. When the PMMA/Ag volume ratio was 2:3, the SERS signal of the molecule reached the strongest. As the silver colloid further addition, the AgNPs accumulate to form large clusters, inhibiting the formation of electromagnetic fields between the particles, resulting in a weakening of the Raman signal. Therefore, the AgNPs@PMMA/P-Si substrates with PMMA/Ag volume ratio of 2:3 were determined as optimal. 

To further investigate the SERS effect of the AgNPs and PMMA composite structures and demonstrate the advantages of this 3D pyramid structure for SERS activity, we analyzed and compared the Raman spectra of R6G (10^−7^ M) molecules adsorbed on pyramid-shaped AgNPs and PMMA composite (P–AgNPs@PMMA), AgNPs attached to the pyramid-shaped PMMA film (P-AgNPs/PMMA) and planar AgNPs and PMMA composite (F–AgNPs@PMMA) flexible SERS substrates in Figure 3c. Compared to F–AgNPs@PMMA, P–AgNPs/PMMA has a higher SERS signal, which indicates that the inclined surface of the 3D pyramid structure could facilitate the aggregation of probe molecules and create high-density 3D “hot spots”. In addition, the AgNPs and PMMA composite structures provide uniform space distribution for analyte detection in designated hot spot zone, avoiding the calculation of Raman enhancement factor error caused by different SERS spectra measured in different regions, achieving reliable enhancement factor measurement. The P–AgNPs@PMMA substrate provides a strong enhancement factor equivalent to P–AgNPs/PMMA substrate, and obtains high spatial uniformity and good stable SERS spectrum. The uniform detection geometry can avoid signal distortion caused by direct contact with the probe molecules to cause deformation of the probe molecules, obtaining a more stable SERS signal. As a whole, the high SERS intensity of the composite P–AgNPs@PMMA substrate benefits from the high-density 3D “hot spots”. More importantly, the hybrid detection geometry ensures uniform distribution of analyte molecules in the designated hot spot zone, and the incident light can penetrate the external PMMA film to trigger the localized plasmon resonance of the encapsulated AgNPs, achieving strong enhancement factor and stable Raman signal.

R6G and CV, most commonly used for SERS molecules, were chosen to test the SERS performance of the P–AgNPs@PMMA substrate (Figure 4). Figure 4a illustrates the Raman intensities of R6G deposited on the P–AgNPs@PMMA substrate at various molecular concentrations. It can be seen that the intensity of SERS peaks decreased with the decreasing concentration of R6G solution. Obviously, the 614 and 774 cm^−1^ peaks of R6G were still observed on the substrate even when the concentration was decreased to 10^−13^ M, indicating that the substrate exhibits strong Raman effects. Here, we selected the 614 cm^−1^ peak to further investigate the relationship between the Raman intensity and R6G concentration. The reasonable linear response of the P–AgNPs@PMMA substrate is displayed in Figure 4b. The correlation coefficient (R^2^) values of the peaks at 614 and 774 cm^−1^ can reach 0.988 and 0.989, respectively. In addition, we have calculated the number of adsorbed molecules at different concentrations. The number of adsorbed molecules are estimated by
(1)$$N=\frac{C_{m}\;V_{m}S_{laser}\;}{S_{s}}N_{A},$$
where *C*_m_ is the concentration of molecular solution, *V*_m_ is the volume of molecular aqueous solution dipped onto the whole substrate, *S*_S_ is the area of the whole substrate, *S*_laser_ is the area of the focal spot of laser, and *N*_A_ is the Avogadro’s constant. In the experiments, 20 μL different concentration R6G and CV aqueous solutions were respectively dipped in the substrate. The diameter of the focused laser spot is ~2 μm. The size of the substrate is 5 mm × 5 mm [23]. We obtained the number of adsorbed molecules in the range of 1.51 × 10^6^ − 0.15, depending on different molecules concentration (10^−6^–10^−13^ M). The number of adsorbed molecules gradually increases with the molecular concentration increases. Linear correlation between concentration and intensity can be supported by the number of adsorbed molecules. These results demonstrate the well linear dependence of the substrate. We calculate the enhancement factors (*EF*) for R6G molecule absorbed on the P–AgNPs@PMMA substrate, using the following standard Equation:(2)$$EF=\frac{I_{SERS}/N_{SERS}}{I_{RS}/N_{RS}}.$$

*I*_SERS_ and I_RS_ are the peak intensity of the SERS spectra and the Raman intensity obtained from the SERS sample and pure PMMA film, respectively. *N*_SERS_ and *N*_RS_ are the numbers of target molecules illuminated by the laser spot on the sample and pure PMMA film respectively [24]. The enhancement factor (**EF**) for 614 cm^−1^ peak of this hybrid structure is calculated as 6.24 × 10^8^. The high EF of the P–AgNPs@PMMA substrate is more superior than the previous metal nanoparticles-based SERS substrates (Table 1), which can be attributed to the combined effect of the 3D-pyramid structure and PMMA film. Pyramid structure demonstrates large specific surfaces, which provide many activity sites, and the inclined surface design of this structure can capture more target molecules. Moreover, the PMMA film acts as a high-dielectric layer and enables the strong surface plasmon coupling between AgNPs embedded in PMMA. The uniform hybrid detection geometry ensures uniform distribution of analyte molecules in the designated hot spot zone, providing reliable enhancement factor measurement, resulting in highly sensitive detection of target molecules. 

To further investigate SERS acitivity of P–AgNPs@PMMA substrate, we also collected the Raman signal of the CV molecule with different concentrations (Figure 4c). It is well-known that the Raman cross-section of CV molecules is significantly different from R6G at the same excitation wavelength [29]. The molecule’s diameter of CV is 15.10 Å [30]. It can be seen that the Raman main peaks located at ~914, 1178, 1533, 1587, and 1620 cm^−1^ are present in the SERS spectra of CV. The two peaks at ~914 and ~1178cm^−1^ are attributed to the ring skeletal vibration of radical orientation. Meanwhile, the peaks around ~1533, ~1587, and ~1620 cm^−1^ are due to the ring C–C stretching modes. The Raman peaks of CV at 914 cm^−1^ can be observed even though the concentration of CV decreased to 10^−12^ M, reaching the minimum detection limit, which can be attributed to the 3D high-density SERS “hot spot”, and the excitation laser can easily penetrate the external PMMA film, causing strong surface plasmon resonance of the embedded AgNPs. Figure 4d shows the Raman intensity of CV changed linearly with the molecule concentration, and the R^2^ value of the peaks at 914 and 1587 cm^−1^ can reach 0.990 and 0.993, respectively. We obtained the number of adsorbed molecules in the range of 1.51 × 10^7^ −1.51, depending on different molecule concentrations (10^−5^–10^−12^ M), which present a good linear fit relationship. These results adequately demonstrate that the P–AgNPs@PMMA substrate possesses perfect SERS performance and good quantitative detection capability.

As a highly sensitive analysis tool, the uniformity and reproducibility of Raman signals are important influence factors for practical applications. We randomly selected 15 spots for statistics on the P–AgNPs@PMMA substrate. Figure 5a shows the SERS signals of the R6G molecules (10^−7^ M) collected on the P–AgNPs@PMMA substrate. It can be seen that the SERS spectra of R6G (10^−7^ M) from different points did not exhibit obvious changes in intensities or shapes. To more intuitively compare the peak fluctuation, the R6G Raman signal intensities at 614, 774, and 1362 cm^−1^ are shown in Figure 5b. These results demonstrate that the intensities of these peaks almost make a horizontal line, and the relative standard deviation of the peaks at 614, 774, and 1362 cm^−1^ are 6.913%, 3.926%, and 2.318%, respectively. These percentages are much lower than the scientific standard (20%), which indicates the high homogeneity and reproducibility of the P–AgNPs@PMMA substrate [31]. This is mainly because the AgNPs and PMMA composite structures provide uniform space distribution for analyte detection in designated hot spot zone, resulting in more intense electromagnetic field coupling, and a more uniform SERS signal. Besides, SERS active substrates not only need high sensitivity, high uniformity, but also good signal stability. Thus, time stability is another important parameter in the SERS detection. We exposed the P–AgNPs@PMMA substrate to the air and performed a SERS test every five days to investigate the intensity change of the peak, as shown in Figure 5c. The Raman signal on the P–AgNPs@PMMA substrate drops very little. Figure 5d shows the corresponding Raman intensities of R6G molecule at 614 cm^−1^. These experimental results demonstrate the P–AgNPs@PMMA hybrid detection geometry not only ensures uniform distribution of analyte molecules in the designated “hot spot” zone, but effectively prevents the embedded AgNPs from being oxidized by reacting with components in the air, as well as protect the target molecules from deformation and signal distortion caused by the direct interaction between AgNPs and molecules. 

Aside from the excellent homogeneity and stability, the P–AgNPs@PMMA substrate also exhibits good mechanical stability (stretching and bending) in practical applications. For tensile testing, the flexible substrate was stretched to ~110%, 120%, 130% (*L*/*L*_0_; *L*: length after stretching; *L*_0_: original length), as shown in Figure 6a. The flexible films were then bent in half with various bending cycles (Figure 6b). The optical image of stretching and bending are presented in Figure 6a_1_, b_1_. The Raman signals were analyzed for 10^−7^ M R6G molecules absorbed on the substrate after each mechanical stimulus (Figure.6a_2_, b_2_). In order to draw comparisons, the relative intensities of the characteristic peaks with error bars (same batch but different positions) at 614 and 774 cm^−1^ were collected to plot histograms for the stretching and bending processes, as displayed in Figure 6a_3,_ b_3_. The histograms suggest that the Raman fingerprint peaks at 614 and 774 cm^−1^ remain mostly unchanged compared to the original state before mechanical stimulation. It can be seen that the SERS substrate can achieve mechanical deformation with little loss of SERS performance, and can better meet the requirements of SERS detection for uneven surfaces. This may be due to the good mechanical resistance of the PMMA film and the stability of the embedded AgNPs. It is well-known that the intensity of SERS is not only related to the distance of gaps, but also the number of molecules between gaps. During mechanical stimuli, the distance between silver nanoparticles becomes larger, which creates a large “hot spot” zone. The density of “hot spot” decreases within a certain incident laser area, while the number of adsorbed molecules increases in the “hot spot” zone. Therefore, mechanical changes in a certain range will not change the Raman intensity of the corresponding target molecule.

During mechanical strain, stretching the P–AgNPs@PMMA substrate yields smaller gaps between neighboring particles, resulting in control over the electromagnetic fields coupling on a nanoscale [32]. To better understand the SERS behaviors of the flexible substrate during tensile testing, we employed commercial COMSOL software to model the local electric field properties of the flexible substrate. Herein, we model the electric field distributions of the SERS film to investigate the enhancement behavior. The height (∼3 μm) and space (∼4 μm) of the P–Si geometry was chosen according to the actual sample. The AgNPs embedded in the PMMA film is modeled as a sphere with a diameter of 78 nm and an interparticle distance of 15 nm, which mirrors the actual size in terms of SEM (Appendix A). Additionally, the incident light is 532 nm, and the refractive index of PMMA is 1.5, according to the actual experiments. Figure 7a–d respectively show the local electric field distributions at the y–z cross-section of the SERS film after stretching to ∼0%, 10%, 20%, and 30%. The bottom left in Figure 7d exhibits the theoretical model of P–AgNPs@PMMA substrate. SEM image of the single pyramid-shaped AgNPs structure is also present in the inset of Figure 7d (bottom right). These results indicate that the electric field near the surface of the pyramid does not obviously change as the stretching length increases. Although some subtle differences have occurred, the plasmon coupling between high-density AgNPs embedded in the PMMA film remains unaffected, allowing for almost constant Raman signals, which is consistent with our experimental results.

To further investigate the practicability of the P–AgNPs@PMMA substrate, the in-situ detection of a solution containing biochemical molecules was performed. Adenosine is the metabolite of adenine nucleotides, which are regarded as major neuromodulators. Furthermore, as the core molecule of ATP and nucleic acids, adenosine forms a unique link among cell energy, gene regulation, and neuronal excitability [18,33]. Here, adenosine molecules were dissolved in deionized water, and the molecular concentration was measured at 10^−8^ M. The substrate was placed on the surface of the prepared adenosine aqueous solution. Next, in-situ detection was conducted, as exhibited in Figure 8a. The optical picture of the in-situ detection of adenosine molecules can be observed in Figure 8b. The detected results are presented in Figure 8c. There were no obvious Raman signals (orange line) obtained without the P–AgNPs@PMMA substrate. However, when the activated surface of the P–AgNPs@PMMA substrate encountered the solution, the characteristic peaks at 729, 1257, and 1329 cm^−1^ of adenosine molecules were easily detected (green line), which was attributed to the active surface of the substrate interacted with probe molecules, enhancing the Raman signal. The experimental results suggest the substrate’s promising application to practical in-situ detection of biochemical molecules. Recently, MB is often used as the fish medicine or the disinfector for the fishpond. If MB molecules have not been completely removed from the skin of fish, it would be harmful to humans’ health. We perform MB molecular detection by swabbing the surface of the fish skin using the P–AgNPs@PMMA substrate shown in Figure 8d. Figure 8e exhibits the SERS spectra of MB molecules. Obviously, the Raman intensity decrease with the decrease of the MB concentration. Based on these particle detection, the P–AgNPs@PMMA substrate has shown great potential in noninvasive and ultrasensitive molecular detention, such as the detecting on the surfaces with any arbitrary morphology and aqueous solution.

## 4. Conclusions

In summary, we proposed a rapid and convenient method for fabricating 3D P-AgNPs@PMMA flexible platforms, which enable highly sensitive single-molecule detection and provide a reproducible and stable Raman signal response. Benefiting from the effective oscillation of light inside the pyramid valley could provide wide distributions of 3D “hot spots” in a large space. The inclined surface design of the pyramid structure could facilitate the aggregation of probe molecules, achieving highly sensitive detection of R6G and CV. The AgNPs and PMMA composite structures provide uniform space distribution for analyte detection in designated hot spot zone, and the incident light can penetrate the external PMMA film to trigger the localized plasmon resonance of the encapsulated AgNPs, achieving enormous enhancement factor (~$6.24\times 10^{8}$). Additionally, the substrate maintains a stable SERS signal under various mechanical stimuli such as stretching and bending. As a practical application of the SERS substrate, we achieved the in-situ Raman detection of adenosine aqueous solution and the MB molecule detection of the skin of a fish. Our experimental results suggest promising application prospects for detection on the surfaces with any arbitrary morphology and aqueous solution.

## Figures and Tables

**Figure 1 polymers-12-00392-f001:**
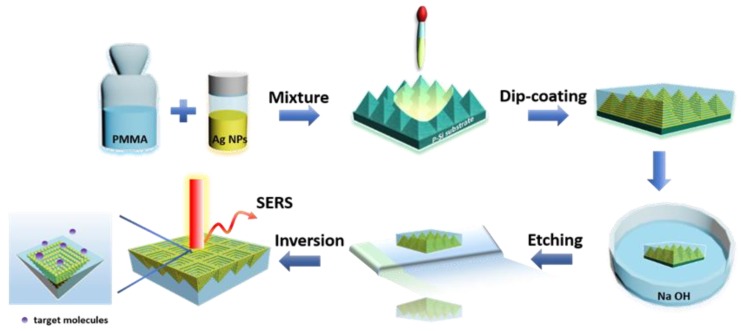
Schematic illustration of the fabrication process for the P–AgNPs @PMMA substrate.

**Figure 2 polymers-12-00392-f002:**
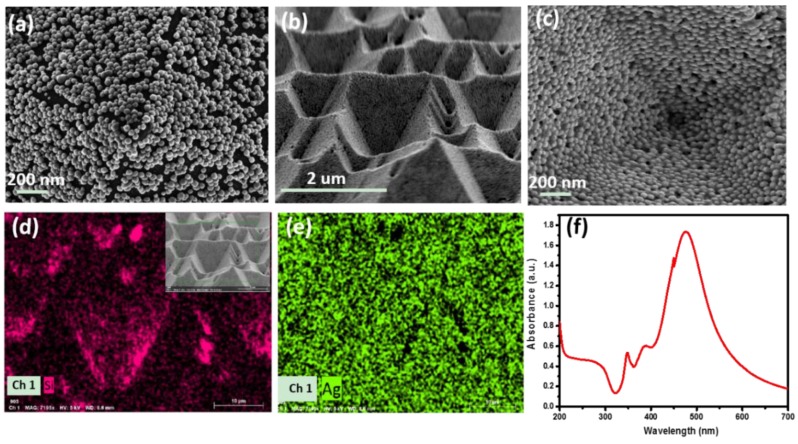
(**a**) Scanning electron microscopy (SEM) image of AgNPs deposited on the P-Si substrate; (**b**) and (**c**) are SEM images of the P–AgNPs@PMMA substrate in different magnification; (**d**) and (**e**) energy dispersive spectrometer (EDS) mapping of the P–AgNPs@ PMMA flexible substrates; (**f**) UV–vis absorption spectra showing the SPR peak of the AgNPs at ~476 nm.

**Figure 3 polymers-12-00392-f003:**
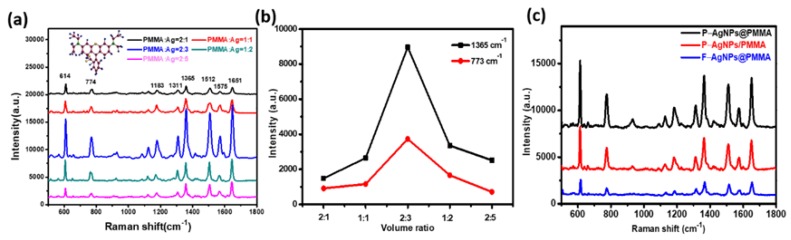
(**a**) The surface-enhanced Raman scattering (SERS) spectra obtained from the P–AgNPs@PMMA substrate with different PMMA/Ag volume ratio; here, the concentration of rhodamine 6G (R6G) is 10^−7^ M. Insert: the structural formula of R6G molecule; (**b**) change in Raman intensity of R6G at 774 and 1365 cm^−1^ as a function of volume ratio; (**c**) SERS spectra of 10^−7^ M R6G absorbed on the P–AgNPs@PMMA (black line), P–AgNPs/PMMA (red line) and F–AgNPs@PMMA (blue line) substrates.

**Figure 4 polymers-12-00392-f004:**
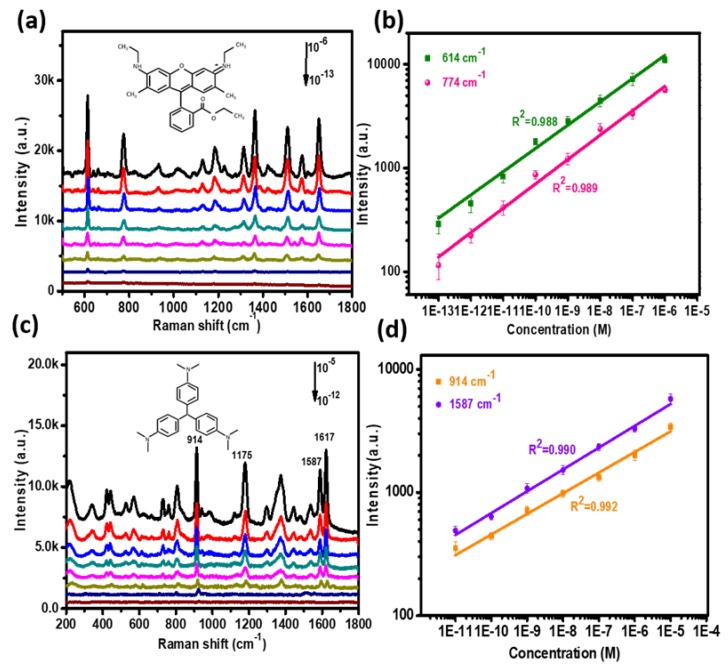
(**a**) Raman spectra of R6G from 10^−6^ to 10^−13^ M on the P–AgNPs@PMMA substrate. Insert: the structural formula of R6G molecule; (**b**) the Raman spectra of R6G at 614 and 774 cm^−1^ as a function of the molecular concentration on the P–AgNPs@PMMA substrate in log scale; (**c**) Raman spectra of CV with concentration from 10^−5^ to 10^−12^ M on the P–AgNPs@PMMA substrate. Insert: the structural formula of CV molecule; (**d**) the Raman spectra of CV at 914 cm^−1^ and 1567 cm^−1^ as a function of the molecular concentration on the P–AgNPs@PMMA substrate in log scale.

**Figure 5 polymers-12-00392-f005:**
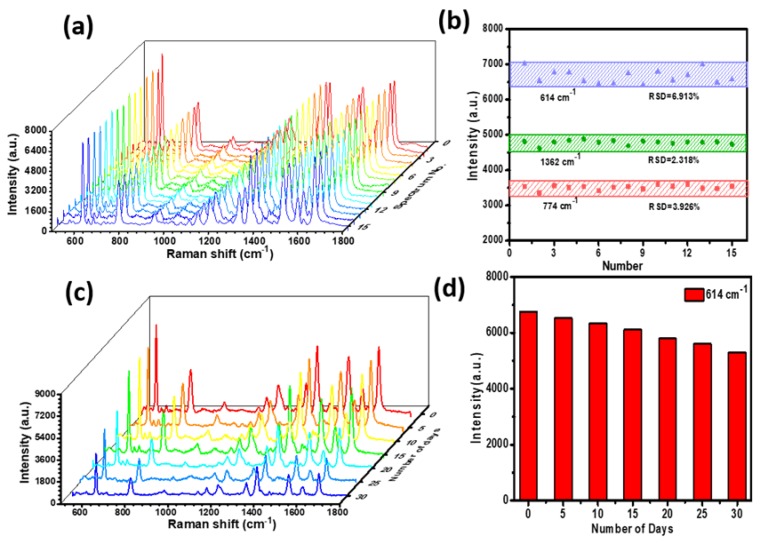
(**a**) Raman spectra collected from different points on the P–AgNPs@PMMA substrate for R6G molecules. The R6G concentration is measured at 10^−7^ M; (**b**) the peaks at 614, 774, and 1362 cm^−1^ demonstrate relative intensities collected from the Raman spectra; (**c**) the Raman spectra of the P–AgNPs@PMMA substrate was measured every five days at room temperature; (**d**) the Raman intensity of the 614 cm^−1^ peaks for R6G from (c).

**Figure 6 polymers-12-00392-f006:**
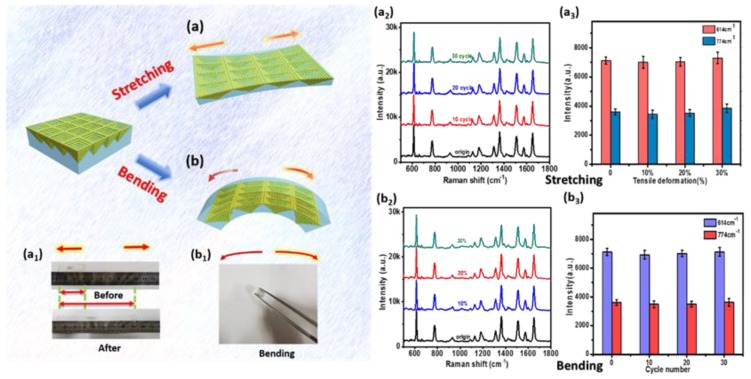
Durability tests with mechanical stimuli of the P–AgNPs@PMMA substrate. Schematic illustration: comparative Raman spectra and the SERS intensity of R6G at 614 and 774 cm^−1^ peaks; (**a**) after stretching the P–AgNPs@PMMA substrate to ~10%, 20%, 30%; (**b**) after the bending the P–AgNPs@PMMA substrate in half; (**a_1_**) the optical image of stretching; (**b_1_**) the optical image of bending; (**a_2_**) raman signals of stretching; (**b_2_**) raman signals of bending; (**a_3_**) histograms for the stretching; (**b_3_**) histograms for the bending.

**Figure 7 polymers-12-00392-f007:**
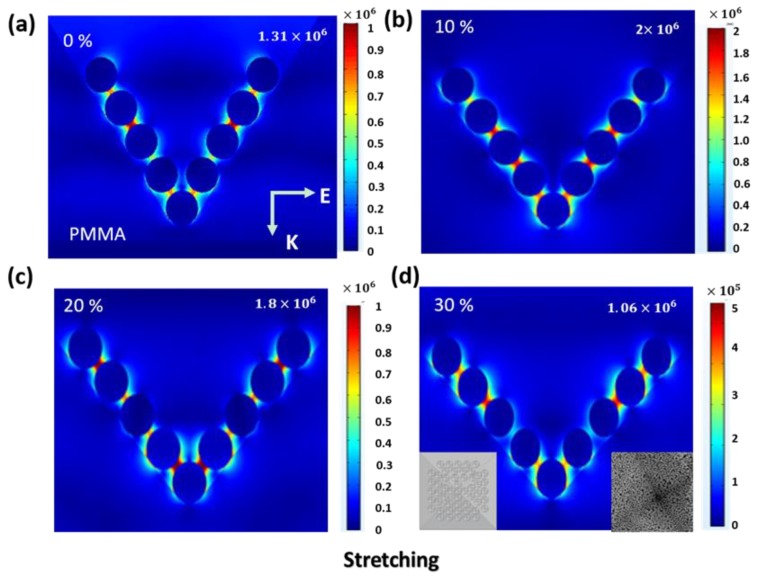
The respective Y–Z views of the electric filed distributed on the P–AgNPs@PMMA substrate after stretching to ~0% (**a**), 10% (**b**), 20% (**c**), and 30% (**d**).

**Figure 8 polymers-12-00392-f008:**
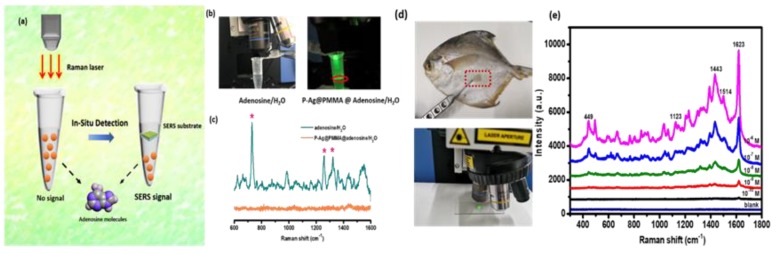
(**a**) Schematic illustration showing the in-situ detection process of the adenosine solution; (**b**) the optical picture of the in-situ detection of adenosine molecules; (**c**) Raman spectra of adenosine molecules before (orange line) and after (green line) placing the P–AgNPs@PMMA substrate on the adenosine solution to confirm the SERS effect. Insert: in-situ detection of adenosine molecule; (**d**) the optical picture of detecting of methylene-blue (MB) by swabbing the marine fish surface; (**e**) SERS spectra of MB obtained by swabbing fingerprint on the marine fish surface.

**Table 1 polymers-12-00392-t001:** Comparing the performance of various flexible SERS sensors.

SERS Substrate	Analytes	Enhancement Factor (EF)	Ref.
Ag-nanosheet-grafted polyamide-nanofibers	4-mercaptobenzoicacid	2.2 × 10^7^	[25]
Ag decorated microstructured PDMS substrate fabricated from Taro leaf	Malachite green	2.06 × 10^5^	[16]
GNS/PDMS	Benzenedithiol	1.9 × 10^8^	[13]
AuNR/fitter paper	1,4-Benzenedithiol	5 × 10^6^	[26]
Flexible free-standing silver nanoparticle-graphene	Rhodamine6G	1.25 × 10^7^	[27]
AgNP/fitter paper by brushing technique	Rhodamine 6G	2.2 × 10^7^	[28]
Flexible AgNP@PMMA/P-Si	Rhodamine 6G	6.24 × 10^8^	This work

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
