# Peer review of "3D Ultrasensitive Polymers-Plasmonic Hybrid Flexible Platform for In-Situ Detection"

_polymers, 2020, doi:10.3390/polym12020392_

Round 1
Reviewer 1 Report
In the manuscript submitted to the Polymers the Authors describe a rapid and convenient method for 3D flexible platforms fabrication based on PMMA and Ag nanoparticles, which enable highly sensitive single molecule detection of rhodamine 6G and crystal violet, providing a reproducible and stable Raman signal response. Additionally, it is shown that the substrate maintains a stable SERS signal under various mechanical stimuli.
Generally, the paper is very well organized, clearly written and with relatively high scientific content. We can find all necessary information including experimental details, description and analysis of experimental results together with suggestions of a very promising application. In my opinion the manuscript can be accepted in the present form.
Author Response
Dear Editors and Referees:
Thank you for your letter and the referees’ comments concerning our manuscript polymers-699871 (“3D Ultrasensitive Polymers-Plasmonic Hybrid Flexible Platform for In-Situ Detection”). Those comments are all valuable and very helpful for revising and improving our paper, as well as the important guiding significance to our researches. We revised the manuscript in accordance with the referees’ comments, and carefully proof-read the manuscript to minimize typographical, grammatical, and bibliographical errors. Revised portions are marked using “Track Changes” function in Microsoft Word. The main corrections in the paper and the responds to the referees’ comments are as flowing:
List of Actions:
LOA1: Figure 2(a) in the original manuscript is transferred to supplementary material.
LOA2: The description of the number of adsorbed molecules have been added in Page 7: Line: 188-199 and Page 8: Line: 229-231.
LOA3: The explanation of changes of density of "hot spot" in the process of mechanical stimulation have been added in Page 9: Line: 281-286.
LOA4: Some funding number has been replaced.
LOA5: The size and position of some pictures in the manuscript have been adjusted reasonably for aesthetics.
LOA6: All typos have been removed.
Response to Reviewer 1 Comments
Special thanks to you for your careful review.
We tried our best to improve the manuscript and made some changes in the manuscript. These changes will not influence the content and framework of the paper. We appreciate for Editors/Referees’ warm work earnestly, and hope that the correction will meet with approval. Once again, thank you very much for your comments and suggestions.
Sincerely yours,
Mei Liu
Meimei Wu
Reviewer 2 Report
This paper reports on preparation of 3D-pyramidal Ag-NPs in PMMA for SERS.
SERS may be an important technique for nano-analysis based on nano-materials. So this topic should be published in Polymers essentially, though some points should be improved before acceptance.
(1) The main point of this study, namely formation of 3D-pyramidal Ag-NPs, should be proved more direct methods. Figure 2(a) may be so weak as a proof.
(2) "Hot spot" should be illustrated visually, for example imposed in Figure 2(e). Moreover, adsorbed molecules and "hot spot" should be compared in their sizes quantitatively.
(3) In Figure 4 (b) and (d), linear correlation between concentration and intensity should be supported by the number of adsorbed molecules actually. Please add effective proofs for such discussion, too.
(4) In Figure 6, can mechanical stimuli change the density of "hot spot" actually? Related to query (2), if construction of Ag-NPs is kept in PMMA films
without gaps, it must be kept and the difference of SERS should be attributed to another reasons. Please exhibit structurally supporting data, too.
That's all.
Round 2
Reviewer 2 Report
This version was well corrected, so it should be accepted as it is.
That's all.